# Impact on step count by commitment-based health application

Masaki Takebayashi[1,2,3], Mira Namba[4]*, Tatsuya Koyama[5], Yudai Kaneda[6], Hiroyuki Kawaguchi[7], Chiaki Uemura[7], Megumi Shibuya[7], Shin Murakami[7], Hiroshi Fukuda[1,8], Hirohide Shibutani[2]

1 The Research Group on Health Education and Promotion, Japan Society for Occupational Health, Tokyo, Japan, 2 Faculty of Sociology, Aomori University, Aomori, Japan, 3 Graduate School of Health Sciences, Aomori University of Health and Welfare, Aomori, Japan, 4 School of Medicine, Keio University, Tokyo, Japan, 5 Faculty of Human Life Sciences, Mimasaka University, Okayama, Japan, 6 School of Medicine, Hokkaido University, Hokkaido, Japan, 7 A10 Lab Inc, Tokyo, Japan, 8 Department of Advanced Preventive Medicine and Health Literacy, Graduate School of Medicine, Juntendo University, Tokyo, Japan

* mirrornamba@keio.jp

**Data Availability Statement:** All relevant data are within the paper and its Supporting information files.

**Funding:** This research was supported by the Shizuoka Prefecture grant (2022-302). The funders

## Abstract

### Objective

Prior research has implied that promoting sustaining physical activity through nudges is challenging and boosting health literacy is important for the long-term establishment of behaviors. This study aimed to investigate the effects of commitment-based health application on step count and health literacy.

### Methods

A control experiment was conducted involving employees from companies located in Shizuoka Prefecture, Japan. Participants were divided into three groups: the commitment app group (utilizing a commitment-based application "Minchalle," where teams of around five members were randomly assigned to declare a target step count and report daily step count with pictures), the self-commitment group (individuals declaring a target step count and endeavoring on their own), and the control group (no intervention). Changes in step count and health literacy were examined over one month.

### Results

A total of 109 employees from 7 companies participated. The changes in step count were an increase of 893 steps for the commitment app group, 243 steps for the self-commitment group, and 178 steps for the control group, with a significant increase in the commitment app group compared to the control group. Regarding health literacy measures, there was significant progress in four items out of five for the commitment app group compared to the control group, and significant progress in one item for the self-commitment group compared to the control group.

had no role in data collection and analysis, decision to publish, or preparation of the manuscript.

**Competing interests:** Hiroyuki Kawaguchi, Chiaki Uemura, Megumi Shibuya and Shin Murakami are employees of A10 Lab Inc. The other authors declare that they have no competing interests. This does not alter our adherence to PLOS ONE policies on sharing data and materials.

## Conclusion

Communication within the app teams, such as commitment, sharing photos of their goal achievements and provide encouraging comments to others, functioned as social nudges, suggesting the potential for an immediate increase in step count and long-term behavioral reinforcement through improved health literacy.

## 1 Introduction

Physical inactivity ranks as the third leading risk factor for mortality among Japanese individuals [1], and promoting physical activity is an urgent public health issue. Japan's health strategy, "Health Japan 21 (the second term)," initiated in 2013, set daily step count goals of 9,000 for men and 8,500 for women, but the final report in 2022 revealed that these targets were not met [2]. Notably, there was no noticeable change in the step count among working-age males, and a decline was observed among females [3]. Moreover, the implementation of remote work due to the impact of COVID-19 has also been cited as potentially contributing to a decrease in step counts [4]. Therefore, effective interventions to increase the working population's step count are urgently needed.

Promoting healthy behavior through nudges has received a lot of attention. Nudge is defined as "any aspect of the choice architecture that alters people's behavior in a predictable way without forbidding any options or significantly changing their economic incentives [5]." An example of a nudge is "commitment," whereby one's future actions are predetermined by declaring behavioral goals, and it appeals to the psychological characteristic of wanting to follow through on something once it has been stated [6]. Commitment falls under the category of Social nudges [7]. Contrary to fewer reports targeting physical activities, a commitment was reported to be most successful when targeting diet [8]. A systematic review on nudges of physical activities shows no commitment study; among 88 studies, 53 were prompting (i.e. prompting taking the stairs through footprints on the floor), 24 were message framing (i.e. emphasizing the benefits of the physical activities), 12 were social comparison (i.e. providing information about others), 8 were feedback (i.e. giving feedback on one's performance,) 1 was default change (placing a desk at stand-up height) and 1 was anchoring (giving a high-level goal) [9]. However, publication bias might prevent reporting commitment to physical activity.

It should be noted that nudging may be effective in immediate behavioral changes, but there is little evidence that nudging results in lasting behavioral changes [10]. Indeed, boosting health literacy is important for the long-term establishment of behaviors [11]. The likelihood of behavior becoming entrenched may increase by taking a step forward with nudges and then boosting it through improved health literacy.

Recently, with the development of remote work, interventions using health applications have been becoming increasingly important [12]. If health applications can be linked to both an increase in step count and improvement in health literacy, it is expected that they would become an effective intervention for promoting physical activity.

This study aimed to investigate the impact of commitment-based health applications on workers' step count and health literacy.

## 2 Methods

### 2.1 Research design, participants and allocation, intervention

A control experiment was conducted with three parallel groups. Recruitment for the companies was conducted through the Shizuoka Prefectural Government, Japan, and recruitment for the participants was conducted through the companies receiving the letter of participation decision to the program. Allocation was performed based on preferences submitted in advance by each company, and companies that did not submit preferences were assigned randomly. The intervention was conducted in January 2023 [T1], and a one-month follow-up was carried out until February [T2]. The inclusion criteria were regular employees of the company, $\geq 20$ years, and the exclusion criteria were those receiving walking-related guidance from medical institutions, and those who failed to submit either of the questionnaires. In the self-commitment and control groups, those using commitment apps were excluded.

**2.1.1 Commitment app group.** A kick-off meeting was held online to explain the purpose of the project and ethical considerations, and the participants were encouraged to set and declare daily step count goals as New Year's resolutions. Subsequently, they were instructed to download the commitment-based health application "Minchalle" (A10 Lab Inc., Tokyo, Japan) on the spot, and its usage was explained [13]. The charge of the app was free. Minchalle was designed to form teams of up to five people to achieve behaviors by setting specific goals and recruiting users with the same goal. Daily, participants were recommended to share photos of their goal achievements under their nicknames within their team and provide encouraging comments to other members. In the commitment app group, those who downloaded the app were randomly assigned to teams using a random number table.

**2.1.2 Self-commitment group.** A kick-off meeting was held in a manner similar to the commitment app group to declare their target number of steps by writing in the chat thread. However, information regarding the commitment app was not provided, and participants were encouraged to work individually toward achieving their goals.

**2.1.3 Control group.** No intervention was conducted.

### 2.2 Survey

T1 and T2 surveys were conducted via Google Forms anonymously, and the data were matched using anonymized IDs.

### 2.3 Outcomes

**2.3.1 Basic attributes.** Basic attributes such as age, gender, smoking habits, weight, and height (for BMI calculation) were collected.

**2.3.2 Outcomes.** Outcomes were changes in step count and health literacy between T1 and T2. Health literacy was assessed by asking participants about their ability to "collect health-related information from various sources," "extract the information he/she wanted," "understand and communicate the obtained information," "consider the credibility of the information," and "make decisions based on the information, specifically in the context of health-related issues [14]. Additionally, participants of commitment app group and self-commitment group were asked about the factors contributing to achieving their goals at T2.

### 2.4 Statistical analysis

Referring to a previous study on promoting behavior by nudges, we assumed an effect size of 0.30, a power of 0.8, and a significant level of 0.05 [15]. A required sample size of 36 in each

group was computed by G*Power version 3.1.9.4 (Heinrich-Heine-Universität, Düsseldorf, Germany).

The significance of primary attributes was examined using the Chi-squared test, Fisher's exact test for nominal scale, and the Kruskal-Wallis test for ordinal scale. Concerning outcomes, the change in step count was analyzed using the Kruskal-Wallis test. In addition, other items were divided into two categories, "progress made" and "others." They were analyzed using Chi-squared or Fisher's exact test. Additionally, to account for between-cluster variability, an analysis based on a linear mixed model was conducted on changes in step count, and an analysis based on logistic mixed models, including a random intercept, was conducted on changes in health literacy. Missing values were excluded from the analysis for each item. Data analysis was conducted using SPSS version 28 (IBM Japan, Ltd., Tokyo), and the significance level was set at $P < 0.05$ (two-sided test). Post hoc testing was corrected using the Bonferroni method (significance level set at $P < 0.05/3 = 0.017$).

## 3 Results

Seven companies (five insurance, one finance, and one pharmaceutical sales) participated in the study, and all were allocated according to their previously expressed preferences. The flow of each group is shown in the Fig 1, and the basic attributes of participants at T1 are displayed in the Table 1, and the outcomes (T2) in the Table 2. Regarding step count, the commitment app group increased by 893 steps, the self-commitment group by 243 steps, and the control group by 178 steps, with a statistically significant increase in the commitment app group compared to the control group. Out of the health literacy measures, the commitment app group made statistically significant progress in 4 items compared to the control group, and the self-commitment group made significant progress in 1 item compared to the control group. In the commitment app group, 19 people (63.3%) responded that "the app helped in achieving the goal," and the reasons (multiple answers allowed) that the app helped in achieving the goal were, in order, "reporting daily challenges" by 11 people, "gaining peers to work with" by 9 people, "encouraging comments from peers" by 6 people, and "automatic chat delivery from the app" by 2 people. The significance did not change between the results with and without adjusting for clustering.

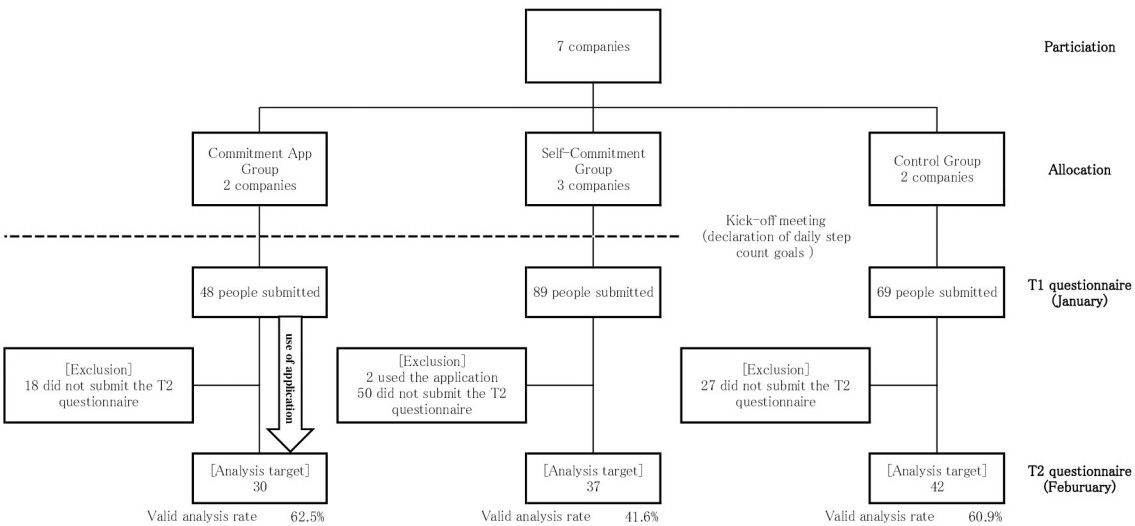

**Fig 1. Trial profile of the three groups.**

**Table 1. Basic attributes at T1.**

| Item content | Commitment App Group (n = 30) | Self-Commitment Group (n = 37) | Control Group (n = 42) | P-value |
|---|---|---|---|---|
| Age (Median, 25th and 75th percentile) | 31 (26, 45) | 32 (28, 47) | 40 (33, 47) | 0.064 |
| Male n(%) | 14(46.7) | 9(24.3) | 22(52.4) | 0.032 * |
| Female n(%) | 16(53.3) | 28(75.7) | 20(47.6) | |
| Smoker n(%) | 8(26.7) | 5(13.5) | 5(11.9) | 0.209 |
| BMI(kg/m2) (Median, 25th and 75th percentile) | 22.1 (20.5, 24.8) | 21.5 (19.5, 23.4) | 21.2 (19.2, 23.9) | 0.311 |
| Step Count (Median, 25th and 75th percentile) | 4999 (3284, 8249) | 4660 (4000, 7480) | 6550 (5000, 7143) | 0.240 |

The median and 25th and 75th percentile values are used for age, BMI, and step count. The Kruskal-Wallis test was performed to examine significance of basic attributes, and the Chi-squared test was performed for n (%).

*The self-commitment group had significantly more female than the control group.

**Table 2. Outcomes (changes from T1 to T2).**

| Item content | Status of improvement | Commitment App Group (n = 30) | Self-Commitment Group (n = 37) | Control Group (n = 42) | P value without adjusting for clustering | P value adjusting for clustering |
|---|---|---|---|---|---|---|
| Changes in step count (Median, 25th and 75th percentile) | − | 893 (230, 1800) | 243 (-1045, 1318) | 178 (-977, 437) | 0.018** | 0.027** |
| Health literacy1 (collect health-related information from various sources) (n, %) | Improved | 11(36.7%) | 6(16.2%) | 0(0%) | <0.001*** | –**** |
| | No change/Worsened | 19(63.3%) | 31(83.8%) | 42(100%) | | |
| Health literacy2 (extract the information he/she wanted) (n, %) | Improved | 8(26.7%) | 9(24.3%) | 4(9.5%) | 0.121 | 0.062 |
| | No change/Worsened | 22(73.3%) | 28(75.7%) | 38(90.5%) | | |
| Health literacy3 (understand and communicate the obtained information) (n, %) | Improved | 11(36.7%) | 8(21.6%) | 4(9.5%) | 0.021** | 0.007** |
| | No change/Worsened | 19(63.3%) | 29(78.4%) | 38(90.5%) | | |
| Health literacy4 (consider the credibility of the information) (n, %) | Improved | 10(33.3%) | 7(18.9%) | 4(9.5%) | 0.041** | 0.015** |
| | No change/Worsened | 20(66.7%) | 30(81.1%) | 38(90.5%) | | |
| Health literacy5 (make decisions based on the information, specifically in the context of health-related issues) (n, %) | Improved | 12(40.0%) | 6(16.2%) | 3(7.3%) | 0.002** | 0.003** |
| | No change/Worsened | 18(60.0%) | 31(83.8%) | 38(92.7%) | | |

n (%) is used except for step count. The Kruskal-Wallis test was used to examine significance for the changes in step count, and a Chi-squared test was used for all others. Both post hoc tests were corrected using the Bonferroni method (P<0.05/3 = 0.17).

**Post hoc tests were corrected using the Bonferroni method (P<0.05/3 = 0.17). Results showed that the commitment app group was significantly higher than the control group.

***Post hoc tests were corrected using the Bonferroni method (P<0.05/3 = 0.17). Results showed that the commitment app group and the self-commitment group were significantly higher than the control group.

**** The P value could not be calculated due to the complex structure of the data.

## 4 Discussion

In this study, the commitment app group showed a statistically significant increase in the number of steps (no significant difference observed in the self-commitment group) and significant progress in 3 items of health literacy (with no progress observed in the self-commitment group) compared with the control group. For Health Literacy 1, "collecting health-related information from various sources," we could not calculate the P value adjusted for clustering. Considering that the P value without adjusting for clustering was < 0.001, the commitment

app group and the self-commitment group might have higher scores than the control group in Health Literacy 1. However, from the viewpoint of eliminating uncertainty, we decided not to mention a significant difference in Health Literacy 1 in this study.

These results suggested that using this health app may contribute to immediate increases in step count and a long-term establishment of behavior through a boost effect caused by improved health literacy. The factors of the app, such as "daily reporting," "gaining peers," and "comments from peers," correspond to social nudge elements such as "self-recording," "peer effect," and "feedback." Previous studies have suggested that self-recording, peer effect, and feedback promote healthy behavior [16–18]. From the results of this study, it was suggested that combining social nudges may lead to an increase in step count.

On the other hand, it was also suggested that commitment alone is insufficient for increasing step count in this study. In the systematic review, the absence of reports on commitment to physical activity is not attributed to publication bias; however, it is considered to be due to the lack of sufficient evidence demonstrating that commitment can effectively promote an increase in the number of steps taken [9].

Progress in "collecting health-related information from various sources," "understanding and communicating the obtained information" and "considering the credibility of the information" could be interpreted as contributions made through communication within the app. The motivation to gather and disseminate credible information with team members, possibly fostered by the peer effect [19], contributed to advancements in health literacy components. Progress in "making decisions based on the information, specifically in the context of health-related issues" appears to be due to setting and declaring goals, but in the self-commitment group, no significant increase was observed. In this study, we anticipated that setting goals and declaring them would be the main factor to increase the number of steps and health literacy. However, it was suggested that what is essential is nudges that promote camaraderie and a series of communication, such as commitment, sharing photos and provide encouraging comments.

Physical activity is an "inter-temporal choice," with the efforts happening now and the health benefits appearing in the future. For individuals who tend to procrastinate on taking action, nudges are useful. However, in the presence of the nudges of physical activities, 68% of the studies reported an effect, whereas after removing the intervention, the effect decreased [9]. Not all workplaces can continuously support physical activity interventions for their employees. Therefore, both immediate step-lengthening nudges and long-term health literacy improvements are essential. Especially with the advancement of remote work, health apps designed with social nudges can be expected as interventions for promoting and sustaining physical activity.

This study has some limitations. First, there were more excluded subjects than anticipated, and as a result, the commitment app group fell short of the sample size. This may have influenced the consideration of significance. Second, the participants might have motivation for physical activity, and the selection bias might occur. Third, this study was a one-month survey in one region, and the results cannot be immediately generalized. Further research is needed to verify the long-term effects and specific health outcomes. Despite these limitations, this study is valuable in suggesting a resolution to the challenge identified in previous research: "Continuing physical activity through nudges is difficult" by communicating with social nudges in the health app.

## 5 Conclusion

Prior research has implied that promoting sustained physical activity through nudges is challenging, and boosting health literacy is essential for the long-term establishment of behaviors.

In contrast, this study suggested that commitment-based health applications may increase step count and health literacy, especially due to communication within the app teams through social nudges.

## Supporting information

**S1 Table. Dataset and results of 109 participants.**
(XLSX)

## Acknowledgments

We sincerely thank the Shizuoka Prefectural Government for their support of our research. We would like to express our gratitude to everyone at the Shizuoka Prefecture Health Promotion Division, Mr. Shota Fujimoto and Ms. Chisako Fukumura of OZMA Nudge Social Design Unit, and all those who participated in the study.

## Author Contributions

**Conceptualization:** Masaki Takebayashi, Chiaki Uemura, Hiroshi Fukuda, Hirohide Shibutani.

**Data curation:** Masaki Takebayashi, Tatsuya Koyama.

**Formal analysis:** Masaki Takebayashi, Tatsuya Koyama.

**Methodology:** Masaki Takebayashi.

**Project administration:** Masaki Takebayashi, Hiroyuki Kawaguchi, Chiaki Uemura, Megumi Shibuya, Shin Murakami.

**Resources:** Hiroyuki Kawaguchi, Chiaki Uemura, Megumi Shibuya, Shin Murakami.

**Writing – original draft:** Masaki Takebayashi, Mira Namba, Yudai Kaneda.

**Writing – review & editing:** Hiroshi Fukuda, Hirohide Shibutani.

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
