## [Decision Letter · Decision Letter 0]

6 Feb 2024

PONE-D-23-34349Impact on Step Count by Commitment-based Health ApplicationPLOS ONE

Dear Dr. Namba,

Thank you for submitting your manuscript to PLOS ONE. After careful consideration, we feel that it has merit but does not fully meet PLOS ONE’s publication criteria as it currently stands. Therefore, we invite you to submit a revised version of the manuscript that addresses the points raised during the review process.

We look forward to receiving your revised manuscript.

Kind regards,

Yasuko Kawahata

Academic Editor

PLOS ONE

Journal Requirements:

2. Thank you for stating the following in the Competing Interests section:"Hiroyuki Kawaguchi, Chiaki Uemura, Megumi Shibuya and Shin Murakami are employees of A10 Lab Inc. The other authors declare that they have no competing interests."

Reviewers' comments:

Reviewer's Responses to Questions

**Comments to the Author**

1. Is the manuscript technically sound, and do the data support the conclusions?

Reviewer #1: Partly

2. Has the statistical analysis been performed appropriately and rigorously? 

Reviewer #1: I Don't Know

3. Have the authors made all data underlying the findings in their manuscript fully available?

Reviewer #1: Yes

4. Is the manuscript presented in an intelligible fashion and written in standard English?

Reviewer #1: No

5. Review Comments to the Author

Reviewer #1: Thanks for the opportunity to review this paper. This study aimed to investigate the effects of commitment-based health application on step count and health literacy. The study addressed an important topic. However, the methods was not clearly stated. Below please find the comments for consideration.

1) Line 89: Please elaborate on previous interventions for promoting physical activities and their effectiveness.

2) Line 92: It was mentioned that "no studies have been reported on the promotion of physical activity through commitment". As such, whether commitment has been applied in other health-promoting behaviors? What are their effectiveness?

3) Line 102: Please provide references.

4) Line 111: What companies were targeted? What were the inclusion and exclusion criteria of participants?

5) Line 115: Please clarify how to allocate the participants into three groups.

6) Line 119: Please describe the content of the App.

7) Line 133: How to make commitments regarding step count?

8) Line 166: Please elaborate on the sample size calculation.

9) Line 169: What were the outcome measures and how to assess the outcomes?

10) Please add conclusion.

6. PLOS authors have the option to publish the peer review history of their article (what does this mean?). If published, this will include your full peer review and any attached files.

Reviewer #1: No

---

## [Author Response · Author response to Decision Letter 0]

5 Mar 2024

We would like to thank the Editors and Reviewers for their time and careful consideration of our manuscript. Please find below a detailed description of the revisions and our responses to the editors and reviewers.

Reviewer 1

Comments to the Author

1) Line 89: Please elaborate on previous interventions for promoting physical activities and their effectiveness.

2) Line 92: It was mentioned that "no studies have been reported on the promotion of physical activity through commitment". As such, whether commitment has been applied in other health-promoting behaviors? What are their effectiveness?

Reply

Thank you for your valuable feedback. We answered 1) and 2) collectively.

We mentioned eating behavior as a commitment to other health behaviors. From the results of a systematic review, we outlined the physical activity promotion interventions and their effects as follows;

Line 82-92 

Contrary to fewer reports targeting physical activities, a commitment was reported to be most successful when targeting diet [8]. A systematic review on nudges of physical activities shows no commitment study; among 88 studies, 53 were prompting (i.g. prompting taking the stairs through footprints on the floor), 24 were message framing (i.g. emphasizing the benefits of the physical activities), 12 were social comparison (i.g. providing information about others), 8 were feedback (i.g. giving feedback on one’s performance,) 1 was default change (placing a desk at stand-up height) and 1 was anchoring (giving a high-level goal) [9]. However, publication bias might prevent reporting commitment to physical activity.

Line 241-245

In the systematic review, the absence of reports on commitment to physical activity is not attributed to publication bias; however, it is considered to be due to the lack of sufficient evidence demonstrating that commitment can effectively promote an increase in the number of steps taken [9].

Line 264-268

However, in the presence of the nudges of physical activities, 68% of the studies reported an effect, whereas after removing the intervention, the effect decreased [9]. Not all workplaces can continuously support physical activity interventions for their employees.

3) Line 102: Please provide references.

Reply

Thank you for your feedback. 

Though we saw a Japanese article about “as face-to-face occupational health programs have diminished during the COVID-19 pandemic,” we cannot deny the possibility that face-to-face programs may be increasing after that. Thus, we changed the expression and added the references.

Line 100-101

Recently, with the development of remote work, interventions using health applications have been becoming increasingly important [12].

12. Schall MC, Jr., Chen P. Evidence-Based Strategies for Improving Occupational Safety and Health Among Teleworkers During and After the Coronavirus Pandemic. Hum Factors. 2022;64(8):1404-11. Epub 20210108. doi: 10.1177/0018720820984583. PubMed PMID: 33415997; PubMed Central PMCID: PMCPMC9282942.

4) Line 111: What companies were targeted? What were the inclusion and exclusion criteria of participants?

5) Line 115: Please clarify how to allocate the participants into three groups.

Reply

Thank you for your kind comments. We answered 4) and 5) collectively.

We mentioned the target companies, the inclusion and exclusion criteria of participants, and how to allocate the participants into three groups.

Line 111-123 

Recruitment for the companies was conducted through the Shizuoka Prefectural Government, Japan, and recruitment for the participants was conducted through the companies receiving the letter of participation decision to the program. Allocation was performed based on preferences submitted in advance by each company, and companies that did not submit preferences were assigned randomly. The intervention was conducted in January 2023 [T1], and a one-month follow-up was carried out until February [T2]. The inclusion criteria were regular employees of the company, ≥ 20 years, and the exclusion criteria were those receiving walking-related guidance from medical institutions, and those who failed to submit either of the questionnaires. In the self-commitment and control groups, those using commitment apps were excluded.

Line 185-187

Seven companies (five insurance, one finance, and one pharmaceutical sales) participated in the study, and all were allocated according to their previously expressed preferences.

6) Line 119: Please describe the content of the App.

Reply

Thank you for your comment. We described the content of the App.

Line 131-137

Minchalle was designed to form teams of up to five people to achieve behaviors by setting specific goals and recruiting users with the same goal. Daily, participants were recommended to share photos of their goal achievements under their nicknames within their team and provide encouraging comments to other members. In the commitment app group, those who downloaded the app were randomly assigned to teams using a random number table.

7) Line 133: How to make commitments regarding step count?

Reply

Thank you for your feedback. We added the method of commitment in the self-commitment group.

Line 139-141

A kick-off meeting was held in a manner similar to the commitment app group to declare their target number of steps by writing in the chat thread.

8) Line 166: Please elaborate on the sample size calculation.

Reply

Thank you for your valuable feedback. Referring to a previous study, we described details of the sample size calculation. 

Line 167-171

Referring to a previous study on promoting behavior by nudges, we assumed an effect size of 0.30, a power of 0.8, and a significant level of 0.05 [15]. A required sample size of 36 in each group was computed by G*Power version 3.1.9.4 (Heinrich-Heine-Universität, Düsseldorf, Germany).

9) Line 169: What were the outcome measures and how to assess the outcomes?

Reply

Thank you for your valuable feedback. We made the "Outcomes section" and specified the outcomes as "change in number of steps" and "change in 5 health literacy scales" in the section. In the "Statistical Analysis section," we added the details of assessing the outcomes.

Line 155-157

2.3.2 Outcomes

Outcomes were changes in step count and health literacy between T1 and T2.

Line 172-182

The significance of primary attributes was examined using the Chi-squared test, Fisher's exact test for nominal scale, and the Kruskal-Wallis test for ordinal scale. Concerning outcomes, the change in step count was analyzed using the Kruskal-Wallis test. In addition, other items were divided into two categories, "progress made" and "others." They were analyzed using Chi-squared or Fisher's exact test. Missing values were excluded from the analysis for each item. Data analysis was conducted using SPSS version 28 (IBM Japan, Ltd., Tokyo), and the significance level was set at P < 0.05 (two-sided test). Post hoc testing was corrected using the Bonferroni method (significance level set at P < 0.05/3 = 0.017).

10) Please add conclusion.

Reply

Thank you for your feedback. We added the conclusion section as follows. 

Line 286-292

5 Conclusion

Prior research has implied that promoting sustained physical activity through nudges is challenging, and boosting health literacy is essential for the long-term establishment of behaviors. In contrast, this study suggested that commitment-based health applications may increase step count and health literacy, especially due to communication within the app teams through social nudges.

---

## [Decision Letter · Decision Letter 1]

5 Jun 2024

Impact on Step Count by Commitment-based Health Application

PONE-D-23-34349R1

Dear Dr. Mira Namba,

We’re pleased to inform you that your manuscript has been judged scientifically suitable for publication and will be formally accepted for publication once it meets all outstanding technical requirements.

Kind regards,

Yasuko Kawahata

Academic Editor

PLOS ONE

Additional Editor Comments (optional):

Reviewers' comments:

Reviewer's Responses to Questions

**Comments to the Author**

1. If the authors have adequately addressed your comments raised in a previous round of review and you feel that this manuscript is now acceptable for publication, you may indicate that here to bypass the “Comments to the Author” section, enter your conflict of interest statement in the “Confidential to Editor” section, and submit your "Accept" recommendation.

Reviewer #1: All comments have been addressed

Reviewer #2: All comments have been addressed

2. Is the manuscript technically sound, and do the data support the conclusions?

Reviewer #1: Yes

Reviewer #2: Yes

3. Has the statistical analysis been performed appropriately and rigorously? 

Reviewer #1: Yes

Reviewer #2: Yes

4. Have the authors made all data underlying the findings in their manuscript fully available?

Reviewer #1: Yes

Reviewer #2: No

5. Is the manuscript presented in an intelligible fashion and written in standard English?

Reviewer #1: Yes

Reviewer #2: Yes

6. Review Comments to the Author

Reviewer #1: Thanks for your revision. All my comments were addressed. I am fine with the revision and do not have additional comments.

Reviewer #2: (No Response)

7. PLOS authors have the option to publish the peer review history of their article (what does this mean?). If published, this will include your full peer review and any attached files.

Reviewer #1: No

Reviewer #2: No

---

## [Editor Report · Acceptance letter]

2 Jul 2024

PONE-D-23-34349R1 

PLOS ONE

Dear Dr. Namba, 

I'm pleased to inform you that your manuscript has been deemed suitable for publication in PLOS ONE. Congratulations! Your manuscript is now being handed over to our production team.

Kind regards, 

on behalf of

Dr. Yasuko Kawahata 

Academic Editor

PLOS ONE